# 2 Hydroxybutyric Acid-Producing Bacteria in Gut Microbiome and *Fusobacterium nucleatum* Regulates 2 Hydroxybutyric Acid Level In Vivo

**DOI:** 10.3390/metabo13030451

**Published:** 2023-03-20

**Authors:** Fujian Qin, Jiankang Li, Tianxiao Mao, Shuo Feng, Jing Li, Maode Lai

**Affiliations:** 1State Key Laboratory of Natural Medicines, School of Basic Medical Sciences and Clinical Pharmacy, China Pharmaceutical University, Nanjing 210009, China; 3120090278@stu.cpu.edu.cn (F.Q.); 1721091045@stu.cpu.edu.cn (T.M.); 2Xi’an Key Laboratory of Stem Cell and Regenerative Medicine, Institute of Medical Research, Northwestern Polytechnical University, Xi’an 710129, China; lijiankang-1025@nwpu.edu.cn; 3School of Life Science and Technology, China Pharmaceutical University, Nanjing 210009, China; 3121030173@stu.cpu.edu.cn; 4The Clinical Metabolomics Center, China Pharmaceutical University, Nanjing 210009, China; 5Research Unit of Intelligence Classification of Tumor Pathology and Precision Therapy, Chinese Academy of Medical Sciences (2019RU042), Key Laboratory of Disease Proteomics of Zhejiang Province, Department of Pathology, Zhejiang University School of Medicine, Hangzhou 310058, China

**Keywords:** 2-hydroxybutyric acid, gut microbiota, HMP, taxonomic identification, *Fusobacterium nucleatum*

## Abstract

2-hydroxybutyric acid (2HB) serves as an important regulatory factor in a variety of diseases. The circulating level of 2HB in serum is significantly higher in multiple diseases, such as cancer and type 2 diabetes (T2D). However, there is currently no systematic study on 2HB-producing bacteria that demonstrates whether gut bacteria contribute to the circulating 2HB pool. To address this question, we used BLASTP to reveal the taxonomic profiling of 2HB-producing bacteria in the human microbiome, which are mainly distributed in the phylum Proteobacteria and Firmicutes. In vitro experiments showed that most gut bacteria (21/32) have at least one path to produce 2HB, which includes Aspartic acid, methionine, threonine, and 2-aminobutyric acid. Particularly, *Fusobacterium nucleatum* has the strongest ability to synthesize 2HB, which is sufficient to alter colon 2HB concentration in mice. Nevertheless, neither antibiotic (ABX) nor *Fusobacterium nucleatum* gavage significantly affected mouse serum 2HB levels during the time course of this study. Taken together, our study presents the profiles of 2HB-producing bacteria and demonstrates that gut microbiota was a major contributor to 2HB concentration in the intestinal lumen but a relatively minor contributor to serum 2HB concentration.

## 1. Introduction

2-hydroxybutyric acid (2HB) is a metabolite biomarker in various diseases. For example, several studies have indicated that 2HB is a major risk factor for insulin resistance and type 2 diabetes (T2D) [1,2,3,4,5]. The underlying biochemical mechanisms may involve increased lipid oxidation and oxidative stress [5]. Furthermore, 2HB is a valid marker of dysregulation associated with cognitive decline [6] and mitochondrial disease [7]. Increased serum 2HB level is often present in lung cancer [8,9], colorectal cancer [10,11], and hepatocellular carcinoma [12,13]. In contrast, 2HB fell significantly during the first three months of hormonal therapy for patients with prostate cancer [14]. Recent research shows that serum 2HB was enriched in COVID-19 patients versus healthy controls [15]. Importantly, circulating 2HB levels are a robust marker of an elevated hepatic cytosolic NADH/NAD^+^ ratio, and this high NADH/NAD^+^ ratio is causally related to hepatic insulin resistance and glucose tolerance [16].

Previous work suggests that of the 11 organs tested, only the liver and skin released 2HB in fasted pigs [17]. It was recently shown that when localized to the nucleus, lactate dehydrogenase A (LDHA) gains a non-canonical enzyme activity to produce 2HB [18]. Notably, LDH has already been identified in several microbial genera, including *Lactobacillus*, *Fusobacterium*, *Clostridium*, and *Desulfovibrio* [19,20,21,22]. Previous research has shown that 2HB concentration in the intestinal lumen of CRC patients was negatively associated with *pyramidobacter piscolens* and *coprobacter fastidiosus* [23]. Intestinal 2HB level was strongly negatively correlated with *Ruminococcus*, *Lachnospiraceae_UCG.001*, and *Lachnospiraceae_UCG.006* in ischemic stroke rats treated with Naomaitong [24]. Additionally, vancomycin pretreatment increased cecum and serum levels of 2HB [25]. The above studies indicate that aside from the host, gut microbiota could also potentially influence 2HB levels. Whereas, to date, no studies have systematically identified 2HB-producing bacteria and whether gut bacteria can directly affect the level of 2HB in serum.

In this study, we identified potential bacteria that can synthesize 2HB in human gut microbiota by BLASTP (v 2.10.1+). These bacteria are mainly distributed in Proteobacteria and Firmicutes. Further in vitro fermentation experiments were conducted to test the ability of representative human gut bacteria to produce 2HB, and the results show that most of them can synthesize 2HB. We then discovered significantly reduced intestinal lumen and fecal 2HB levels in antibiotic-treated mice. In contrast, oral gavage of *Fusobacterium nucleatum* was effective in augmenting 2HB levels in colon contents. However, serum concentrations of 2HB did not differ significantly in ABX or *Fusobacterium nucleatum* treatment group. In conclusion, this study is the first systematic study of 2HB-producing bacteria and provides important insights into the relationship between intestinal bacteria and the level of 2HB.

## 2. Materials and Methods

### 2.1. Chemicals

Antibiotic cocktails for preparation of pseudo sterile mice: ampicillin sodium (Aladdin, CAS: 69-52-3), metronidazole (Shanghai yuan ye, CAS: 443-48-1), vancomycin hydrochloride (Shanghai yuan ye, CAS: 1404-93-9), neomycin sulfate (Shanghai yuan ye, CAS: 1405-10-3), gentamicin sulfate (Aladdin, CAS: 1405-41-0).

B vitamins for preparation of culture media: folic acid (Macklin, CAS: 59-30-3), pyridoxine hydrochloride (Macklin, CAS: 58-56-0), riboflavin (Macklin, CAS: 83-88-5), biotin (Macklin, CAS: 58-85-5), thiamine (Macklin, CAS: 59-43-8), nicotinic acid (Macklin, CAS: 59-67-6), calcium pantothenate (Macklin, CAS: 137-08-6), Vitamin B12 (Macklin, CAS: 68-19-9), p-aminobenzoic acid (Macklin, CAS: 150-13-0), thioctic acid (Macklin, CAS: 62-46-4).

Standards, internal standards, and derivatization reagents used for 2HB quantification: Sodium DL-2-Hydroxybutyrate (TCI Shanghai, CAS 5094-24-6), (2,3,3-2H3)-Sodium(±)-2-Hydroxybutyrate (ZZBIO, CAS: 1219798-97-6), 3-Nitrophenylhydrazine hydrochloride (Aladdin, CAS: 636-95-3), 1-(3-Dimethylaminopropyl)-3-ethylcarbodiimide hydrochloride (Macklin, CAS: 1892-57-5).

### 2.2. Sequence and BLASTP of Enzyme

It is generally believed that the more similar the protein sequence is, the more similar its structure is, and it may have similar functions. BLASTP is a protein primary structure sequence alignment program developed and managed by NCBI (National Center for Biotechnology Information). The input protein sequence can be compared with the known sequence in the database to obtain the sequence similarity. In this study, sequence acquisition and comparison were essential, as described previously [26]. Briefly, the query enzyme sequences were manually collected from the Refseq database of NCBI. After filtering by peptide length, all sequences were compared against reference sequences from the HMP database (https://www.hmpdacc.org, accessed on 2 March 2021). Performing a BLASTP search and applying a cutoff of either 45% or 40% (LDH) for sequence identity along with an e-value threshold of 1 × 10^−5^ (Figure 1).

### 2.3. Bacterial Strains and Culture Conditions

Bacterial strains were purchased from the American Tissue Culture Collection (ATCC, Manassas, VA, USA), DSMZ, JCM, BNCC, or biobw. Detailed information on the bacteria strains used in this study is provided in Appendix A. Aerobic bacteria were cultured at 37 °C in a liquid medium (MPYG, FT, BHI) or in a medium solidified by the addition of 1.5% agar, whereas anaerobic bacteria were cultured under strictly anaerobic conditions (80% N_2_, 10% H_2_, 10% CO_2_) in an anaerobic chamber. The colony-forming units (CFU) of bacteria were estimated in a spectrophotometer by recording the absorbance at 600 nm. Bacterial strains were identified based on the sequence of their 16S rRNA gene. In short, 16S sequences were amplified using universal PCR primers: 27F (AGAGTTTGATCCTGGCTCAG) and 1492R (GGTTACCTTGTTACGACTT), and the PCR product was subsequently sequenced at Sangon Biotech (Shanghai, China). 

### 2.4. Fermentation Studies

To resuscitate the strains from glycerol stocks, they were streaked onto plates, and single colonies were chosen to inoculate a liquid medium. The cultures were then incubated at 37 °C for 24 h, and this growth was used to establish the first seed culture. Stocks solution of the individual substrate (Asp, Met, Thr, and 2AB) by dissolving them in PBS at 20 mg/mL. Then filter and sterilize using a 0.22 µm filter. For determination of 2HB production, bacteria were inoculated at 4% of the total volume in flasks and cultured for 24 h at 37 °C, and then cultured in the presence of substrate (950 µL bacterial culture plus 50 µL stocks solution of the substrate) solution at a final concentration of 1 mg/mL for another 4 h. 950 µL blank medium plus 50 µL PBS was used as a blank control to test the ability to synthesize 2HB, and 950 µL bacterial culture medium plus 50 µL PBS was used as background control to screen the substrate preference of bacteria.

### 2.5. Animal Experiments

SPF female C57BL/6J mice (5 weeks) were purchased from GemPharmatech Co. Ltd. (Nanjing, China). All animals were housed in a 12 h/12 h light-dark cycle environment and had full access to food and water. After 1 week of adaptation, mice were randomly divided into different groups. 

ABX mice were prepared following the same method as before [27]. Briefly, mice were treated with a combination of various antibiotics (Ampicillin, 5 mg/mL; Gentamicin, 5 mg/mL; Neomycin, 5 mg/mL; Metronidazole, 5 mg/mL; Vancomycin, 2.5 mg/mL) via oral gavage at 0.2 mL per day. After 7 days, serum and fecal were dissected from control and antibiotic-treated (ABX) mice for 2HB quantitative. Meanwhile, fecal samples were also collected, resuspended in sterile water, filtered with a 70 µm strainer, and applied on antibiotic-free agar plates prepared with brain heart infusion (BHI) medium. Both anaerobic (37 °C for 3 days) and aerobic cultures (37 °C for 2 days) were performed to confirm microbiota depletion [28].

For cultured bacteria transfer, *Fusobacterium nucleatum* (Fn) was resuspended in PBS and orally administered doses of 2 × 10^8^ CFU per mouse. Specifically, the 2HB group mice were treated with 100 mg/kg 2HB in 0.2 mL of PBS by oral gavage daily; the Fn group mice received 2 × 108 Fn per mouse daily; the Fn plus Met group mice that received 2 × 10^8^ Fn and 100 mg/kg methionine per mouse daily; the Fn plus Thr group mice that received 2 × 10^8^ Fn and 100 mg/kg Threonine per mouse daily; the Fn plus 2AB group mice that received 2 × 10^8^ Fn and 100 mg/kg 2-aminobutyric acid per mouse daily; the control group received only the carrier, PBS, by gavage in an equal volume. 

All the protocols for animal experiments were reviewed and approved by the Animal Ethics Committee of China Pharmaceutical University (27 January 2022, Approval Code: 2022-01-027).

### 2.6. Quantitative PCR Analysis for Fusobacterium Nucleatum Abundance

Fn quantification was performed as described before. Briefly, genomic DNA (gDNA) was extracted from frozen mice feces samples using the QIAamp Fast DNA Stool Kit (51604, QIAGEN, Hilden, Germany), according to the manufacturer’s instructors [29,30]. Fn quantification was performed by using ChamQ SYBR qPCR Master Mix (Vazyme) on a QuantStudio3 qPCR machine. Relative abundance was calculated by the 2^−ΔCt^ method. Universal Eubacteria (Eu)16S was used as a reference gene. The following primer sets were used: gDNA from mouse stool was examined.
Fn-F: 5′-CAACCATTACTTTAACTCTACCATGTTCA-3′,Fn-R: 5′-GTTGACTTTACAGAAGGAGAT TATGTAAAAATC-3′,Eu-F: 5′-CGGCAACGAGCGCAACCC-3′,Eu-R: 5′-CCATTGTAGCACGTGTGTAGCC-3′.

### 2.7. Publicly Available Metagenomic Sequence Data

The relative abundance of the top 20 genera obtains from publicly available metagenomic sequence data. The metagenomic sequence data of individuals were collected from European Nucleotide Archive (https://www.ebi.ac.uk/ena/browser/view/ (accessed on 5 May 2021), including the populations from Austria (PRJEB7774), Australia (PRJEB7774), China (PRJEB21528, PRJEB10878, PRJNA422434, PRJEB6337, PRJEB12123, PRJEB13870, PRJNA422434, PRJNA422434), Germany (PRJEB27928, PRJEB6070), Denmark (PRJEB2054), France (PRJEB6070), Tanzania (278393), Italy (PRJNA278393), Japan (PRJDB3601), Republic of Korea (PRJEB1690), Peru (PRJNA268964), Sweden (PRJEB1786) and USA (PRJNA177201, PRJEB12449, PRJNA275349, PRJNA48479, PRJNA389280, PRJNA398089, PRJNA268964)

### 2.8. 2HB Quantitation by Liquid Chromatography-Mass Spectrometry (QQQ LC/MS)

Quantitative analysis of 2HB was performed with an Agilent 6495 mass spectrometer (Agilent Technologies, Santa Clara, CA, USA) in negative mode, and multiple reaction monitoring (MRM) was used. For the analysis, a Waters BEH Amide column was employed (50 mm × 2.1 mm inner Q16 diameter, 1.7-mm particle size). The system received 2 μL of the sample through an autosampler, which had been conditioned at 4 °C, while the column temperature was held at 40 °C. An isocratic flow of 85% mobile phase A (0.1% formic acid in water) and 15% mobile phase B (0.1% formic acid in acetonitrile) at a flow rate of 0.3 mL/min was used. The ion source parameters were set as follows: the gas temperature at 300 °C; gas flow at 10 L/min; sheath gas temperature at 350 °C; sheath gas flow at 11 L/min, capillary at 3000 V, and nozzle voltage at 1000 V.

Serum and the supernatant of the bacterial culture were extracted with three times the volume of acetonitrile. Fecal samples (mg) were extracted with ten times the volume of 50% acetonitrile in water (μL). After centrifugation at 12,000 rpm 10 min 4 °C, the supernatant was collected, followed by derivatization as described earlier [31]. Briefly, 40 μL of supernatant, 20 μL 150 mM 3-NPH, and 20 μL 240 mM EDC were added to a 1.5 mL EP tube and sealed with parafilm. Next, a brief vortex of the mixture was for 1 min, and samples were derivatized at 30 °C 60 rpm for 40 min. After the derivatization, an additional 420 μL of 10% acetonitrile was added to the sample, vortexed, centrifuged, and the 80 μL supernatant was transferred to the glass autosampler vial for quantitation.

The quantitation of 2HB was performed concerning their corresponding isotope-labeled internal standards (2HB-D2). The *m*/*z* of monitored ions are as follows: 241/152 (2HB-D2), 239/152 (2HB). Collision energies were 15 for 2HB-D2 and 14 for 2HB. Calibration curves of 2HB were drafted with 2HB standards for absolute quantitation of the biological concentration of 2HB in samples.

### 2.9. Statistical Analysis

The graphical abstract is produced by Figdraw. Data are expressed as mean ± s.e.m. Statistical analysis was performed via either R version 4.0 or GraphPad Prism 8. Pairwise comparisons were conducted using the two-tailed paired Student’s *t*-test, with a significance threshold of *p* < 0.05. 

## 3. Results

### 3.1. Identification of 2HB-Producing Bacteria in Human Gut Bacteria

As an important endogenous metabolite, 2HB-producing bacteria have not been systematically studied, and in this study, we first identified 2HB-producing bacteria in the human microbiome. The proposed biosynthetic pathway of 2HB in gut bacteria was conducted according to the KEGG database (www.genome.jp/kegg, accessed on 10 February 2021) (Figure 1A). As shown in Figure 1A, aspartate, methionine, threonine, and 2-aminobutyric can be used as substrates by bacteria to synthesize 2-ketobutyric acid, a precursor of 2HB, which can be catalyzed by LDH to generate 2HB. We collected synthetic enzymes in the 2HB pathway from the NCBI Reference Sequence Database and compared them with the HMP database (Appendix A and Appendix A). The results showed that enzymes related to 2HB synthesis are widely distributed in the gut microbiota. Specifically, 373,366,550,836 species can use Aspartic acid (Asp), Methionine (Met), Threonine (Thr), and 2-aminobutyric acid (2AB) for the synthesis of 2HB, respectively (Figure 1B–E). Furthermore, these bacteria were mainly distributed in Proteobacteria, Firmicutes, Bacteroidetes, Actinobacteria, Cyanobacteria, an) d Fusobacteria (Appendix A). And the top 20 most abundant genera have the potential to use at least one of Asp, Met, Thr, and 2AB to synthesize 2HB (Figure 1F).

### 3.2. Determination of 2HB Biosynthesis In Vitro

We next tested the ability of 32 strains of bacteria to produce 2HB in vitro by liquid chromatography-mass spectrometry/mass (LC-MS/MS). After 24 h of growth (*A. muciniphila* were cultured for 72 h). 2HB content in supernatant significantly increased in 21 bacterial strains, reduced in 4 bacterial strains, and 7 bacterial strains had no significant changes (Figure 2A and Appendix A). Although 21 strains of bacteria have the ability to synthesize 2HB, their yields vary greatly. Moreover, different bacteria strains have different substrate preferences. For example, *Fusobacterium nucleatum* performs the highest synthetic ability and preferentially uses threonine as the substrate for 2HB production. (Figure 2B and Appendix A).

### 3.3. Gut Bacteria in Mice Are Significantly Positively Correlated with Intestinal Lumen 2HB Level

To determine whether gut bacteria are associated with 2HB levels in C57BL/6J mice, a cocktail of antibiotics (ABX) was used to remove gut bacteria [27], and the successful elimination of symbiotic bacteria was verified by colony loss (Figure 3A,B). ABX treatment did not significantly affect food intake or body weight (Figure 3C,D, and Appendix A). 2HB levels were significantly decreased in the intestinal lumen and fecal, but no significant changes were observed in the serum in ABX-treated mice compared to mice with sterile water treatment (Figure 3E–I and Appendix A). Therefore, these data indicate that gut bacteria in C57BL/6J mice is directly linked to the host intestinal lumen and fecal 2HB level.

### 3.4. Fn Elevates 2HB in the Colon

The in vitro experiments shown in Figure 2 indicate that *Fusobacterium nucleatum* (Fn) has a significantly higher capacity to synthesize 2HB than other bacteria, which use Met, Thr, or 2AB as substrates. Therefore, we investigated whether Fn can affect the 2HB levels in vivo (Figure 4A). With Fn or Fn plus substrate gavage, we observed a marked increase of Fn in feces through qRT-PCR (Figure 4B and Appendix A). Nevertheless, there is no significant change in serum 2HB concentration after 2 weeks of continuous oral gavage of Fn (Figure 4C and Appendix A). 2HB levels were significantly increased in the cecum and colon content of 2HB-treatment versus control mice (Figure 4E,F). In Fn, Fn + Thr, and Fn + 2AB groups, we observed elevated levels of 2HB in the colon only (Figure 4D–G and Appendix A). Altogether, our results suggest that 2HB-producing bacteria can affect the concentration of 2HB locally in the intestinal tract but have no significant effect on serum. 

## 4. Discussion and Conclusions

Multiple evidence suggests that numerous gut microbes metabolites may contribute to host metabolism, such as trimethylamine (TMA) [32], inosine [33], indole propionic acid (IPA) [34], phenylacetylglutamine(PAGln) [35], N,N,N-trimethyl-5-aminovaleric acid (TMAVA) [36]. Previous research has shown that, in addition to endogenous sources, 2HB could be produced by bacteria via lactate dehydrogenase (LDH) enzymes [19,21]. Therefore, we guess that gut bacteria may contribute to 2HB metabolism. Yet, there exists no systematic study on 2HB-producing bacteria. An initial objective of the study was to identify the taxonomic and abundance profiling of 2HB-producing bacteria in the human gut microbiome. We found four pathways for 2HB synthesis in the KEGG database and identified the distribution of pathway enzymes in HMP by BLASTP. Our results have demonstrated that potential 2HB-producing bacteria are ubiquitous in the human gut microbiome (Figure 1). Furthermore, bacteria-carrying enzymes related to 2HB synthesis are mainly distributed in Proteobacteria, Firmicutes, Bacteroidetes, Actinobacteria, and Cyanobacteria in the genus level (Appendix A). 

Consistent with our BLASTP result, in vitro fermentation experiments suggest that most bacteria can synthesize 2HB, although yields vary widely (Figure 2A). What’s more, our data phenocopy findings that different bacterial have different preferred substrates (Figure 2B). The most substantial increase occurred in *Fusobacterium nucleatum* supernatant by about 281 times compared with the blank media. Earlier studies have demonstrated that LDH of *Fusobacterium nucleatum* uniquely exhibits a broad substrate preference for 2-ketobutyric acid (2-KB), a key precursor for the synthesis of 2HB [37]. In addition, *Clostridium sporogenes* and *Peptostreptococcus anaerobius* also have a strong ability to promote 2HB synthesis, which increased by more than ten times in their culture supernatant. Interestingly, *Lactobacillus* and *Bifidobacterium*, which are known for producing lactic acid via LDH, have a weak ability to produce 2HB. These findings collectively intimated that gut microbiota might play an essential role in contributing to the 2HB level.

Here, we demonstrate that ABX-mediated microbiota depletion reduces the accumulation of 2HB in the intestinal lumen and fecal (Figure 3F–I), suggesting a potential direct effect of gut bacteria on 2HB in vivo. By contrast, colon 2HB concentrations are significantly increased after 2HB, *Fusobacterium nucleatum*, and *Fusobacterium nucleatum* plus Thr or 2AB gavage (Figure 4F). The concentration of 2HB in the control group in Figure 3 and Figure 4 showed a gradual increase along the digestive tract, which was consistent with the change in gut bacterial abundance. Strikingly, neither ABX nor *Fusobacterium nucleatum* treatment influenced serum 2HB (Figure 3E and Figure 4C). The concentration of 2HB in the intestinal lumen may be too low to cause a significant change in the serum 2HB pool. Thus, we speculate that the main origin of 2-HB may be derived from the host metabolism. Aside from its potential role as an early biomarker of disease [38], 2HB has also demonstrated high potential to be an important regulatory factor. Earlier research shows that 2HB derived from commensal bacteria ameliorated acetaminophen-induced cell damage and liver injury in mice [25]. Previous studies indicated that 2HB acts as an important antioxidant metabolite for HPV-induced cervical tumor growth [18]. Interestingly, *Fusobacterium nucleatum* [30,39,40,41,42] and *Peptostreptococcus anaerobius* [43,44] have also been implicated in the development of colorectal cancer. Moreover, Studies have shown that supplementation with 2-KB extends the lifespan in wild-type worms [45]. Elderly mice fed with autophagy-induced metabolite 2-KB can prevent hair loss [46]. Collectively, 2HB-producing bacteria may represent a useful therapeutic target in a diverse range of human diseases.

The present study has several limitations. First of all, our research was carried out in normal mice, but whether similar effects will be observed in the state of the disease remains to be evaluated. Because the intestinal barrier may be damaged under the condition of disease, which may help 2HB enter the blood. Secondly, given the positive correlation between the level of 2HB and insulin resistance [16], which is a risk factor for type 2 diabetes and cancer patients, the role of 2HB and 2HB-producing bacteria in disease deserves further study. Finally, although 32 strains of bacteria have been studied in this study, they are still far from reflecting the real bacterial situation in vivo. Even so, this study reveals, for the first time, the potential 2HB-producing bacteria in the human gut microbiome.

In conclusion, we identified the taxonomic profiling of 2HB-producing bacteria in the human gut microbiome. Furthermore, we demonstrated that intestinal bacteria could significantly affect the content of 2HB in the intestinal lumen and feces but have no significant effect on the concentration in serum.

## Figures and Tables

**Figure 1 metabolites-13-00451-f001:**
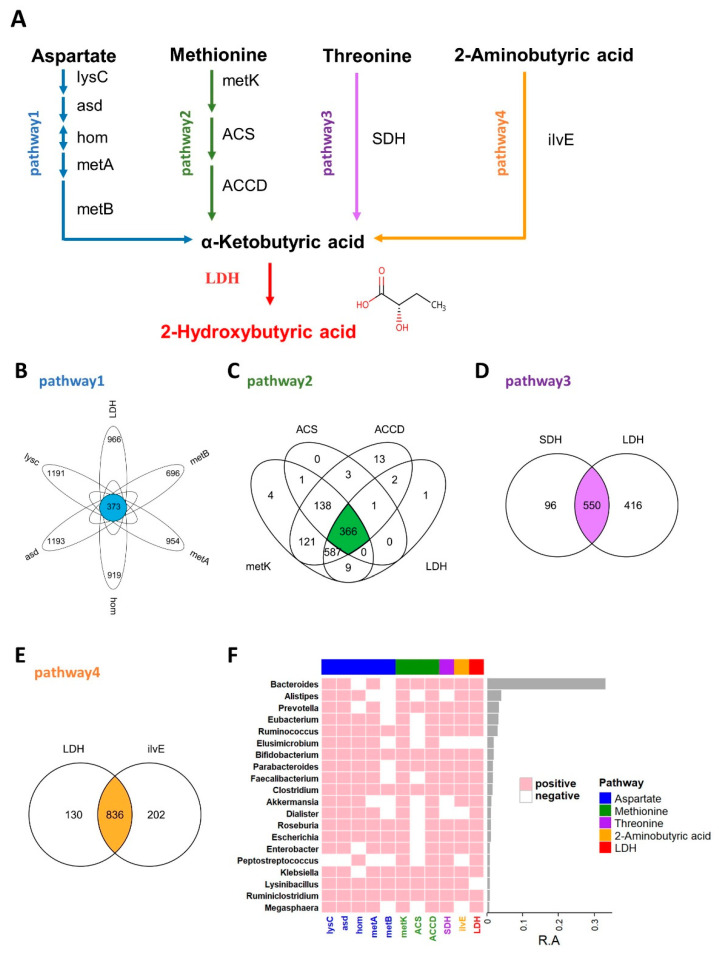
2HB synthesis pathway and the distribution of related enzymes in human gut microbiota. (**A**) Biosynthesis pathway for 2HB production utilizing Aspartate, Methionine, Threonine, and 2-Aminobutyric acid as the substrate. (**B**–**E**) Venn diagram shows the common genes between species in Aspartate, Methionine, Threonine, and 2-Aminobutyric acid pathway, respectively. (**F**) Heatmap of the 20 most prevalent genera in human gut microbiota. Each column represents an enzyme, each row represents a genus, and the positive represents this genus contains enzymes. The relative abundances of the top 20 genera are indicated by the barplot. lysC: aspartate kinase, asd: aspartate-semialdehyde dehydrogenase, hom: homoserine dehydrogenase, metA: homoserine O-succinyltransferase/O-acetyltransferase, metB: cystathionine gamma-synthase, metK: adenosylmethionine synthetase, ACS:1-aminocyclopropane-1-carboxylate synthase 1/2/6, ACCD: 1-aminocyclopropane-1-carboxylate deaminase, SDH: L-serine/L-threonine ammonia-lyase, ilvE: branched-chain amino acid aminotransferase, LDH: L-Lactate dehydrogenase & D-Lactate dehydrogenase.

**Figure 2 metabolites-13-00451-f002:**
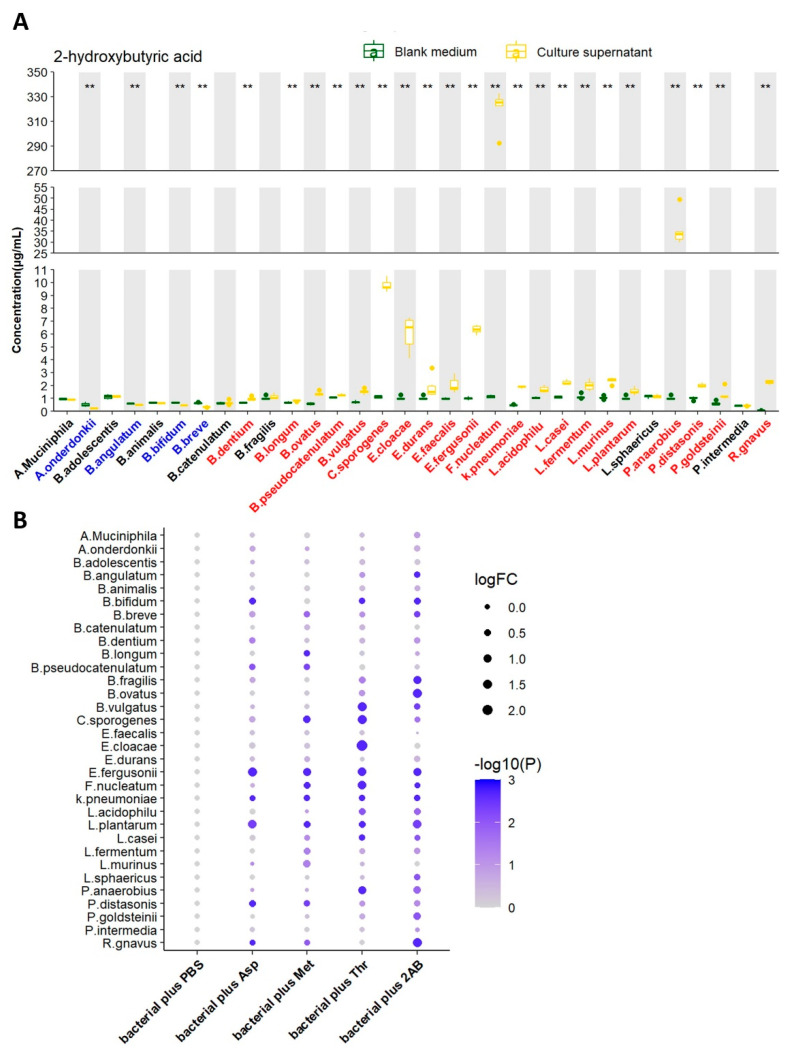
Screening of 2HB-producing bacteria and their preferred substrates in vitro. (**A**) 2HB concentration change from in vitro cultures of 32 strains. Bacteria were colored as follows: blue, 2HB-consuming bacteria; red, 2HB-producing bacteria. (**B**) 2HB concentration change from in vitro cultures of 32 strains that were provided with 1 mg/mL Asp, Met, Thr, or 2−Aminobutyric acid, respectively. The fold change of each bacterial plus substrate was normalized to their own control group (plus an equal volume of PBS), respectively. Further details are provided in the Methods. Asp: Aspartic acid; Met: Methionine; Thr: Threonine; 2AB: 2−Aminobutyric acid. ** *p*  <  0.01.

**Figure 3 metabolites-13-00451-f003:**
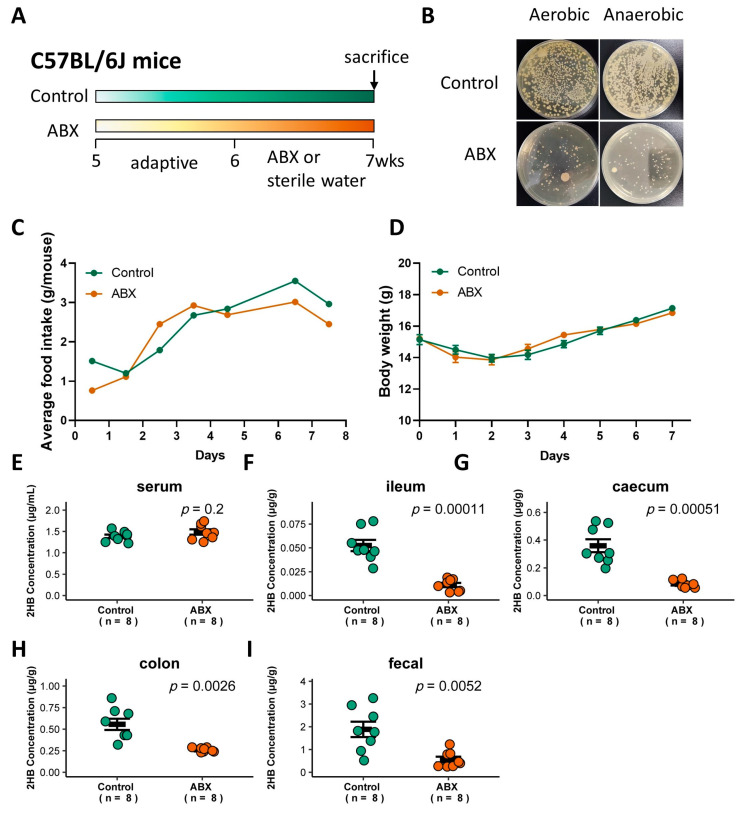
Effect of gavage with ABX on the gut bacteria abundance and 2HB levels of C57BL/6J mice. (**A**) Schematic of the schedule for antibiotic (ABX) administration. Mice received 200 µL of the antibiotic mix (Ampicillin, 5 mg/mL final; Gentamicin, 5 mg/mL final; Neomycin, 5 mg/mL final; Metronidazole, 5 mg/mL final; Vancomycin, 2.5 mg/mL final) by oral for 7 consecutive days. After 1 week, animals were killed, and serum and fecal samples were collected for analysis. (**B**) Fecal bacteria after antibiotic treatment. (**C**,**D**) Average food intake (**C**) and weight change (**D**) in mice with ABX pretreatment for 7 days. (**E**–**I**) Levels of 2HB in serum (**E**), ileum content (**F**), caecum content (**G**), colon content (**H**), and fecal (**I**) of mice after gavage with ABX (*n* = 8).

**Figure 4 metabolites-13-00451-f004:**
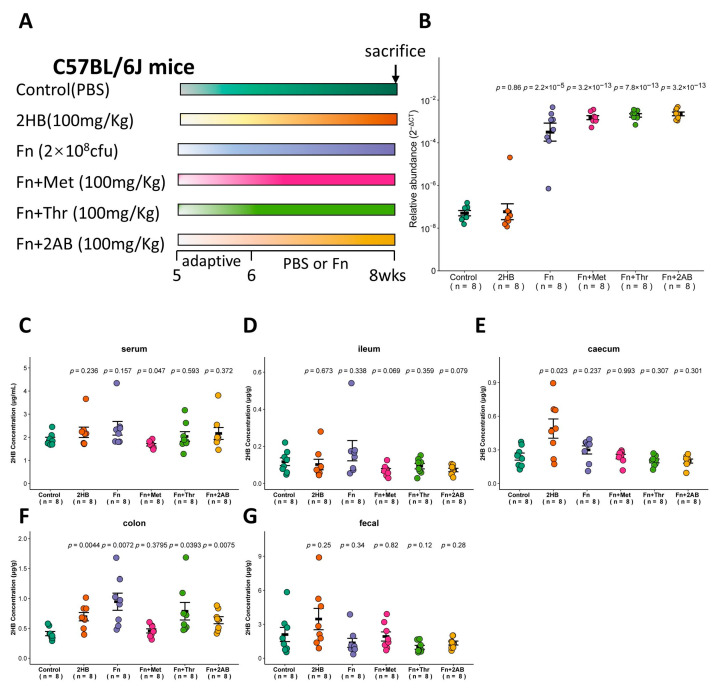
Changes of 2HB levels in C57BL/6J mice after 2 weeks of gavage-feeding *Fusobacterium nucleatum*. (**A**) Schematic overview of experimental design. Mice were gavaged with PBS, *Fusobacterium nucleatum* (Fn), Fn plus Met, Fn plus Thr, and Fn plus 2AB (*n* = 8). At the necropsy on week 8, serum, intestinal contents, and fecal samples were collected for analysis. (**B**) Fn colonization was determined after gavage by qRT-PCR in fecal. (**C**–**G**) Levels of 2HB in serum (**C**), ileum content (**D**), caecum content (**E**), colon content (**F**), and fecal (**G**) of mice colonized with Fn (*n* = 8).

## Data Availability

The authors confirm that the data supporting the findings of this study are available within the article and its Appendix A.

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
