# Peer review of "2 Hydroxybutyric Acid-Producing Bacteria in Gut Microbiome and Fusobacterium nucleatum Regulates 2 Hydroxybutyric Acid Level In Vivo"

_metabolites, 2023, doi:10.3390/metabo13030451_

Round 1

Reviewer 1 Report

There are very interesting aspects to this manuscript while it presented novelty by revealing the relationship between gut microbiome and its influence on 2HB level in mice. The only concern is authors revealed serum 2HB levels are increased in a cohort of colorectal cancer patients but no experiments done with CRC mice, making a weak linkage between their findings in mice and clinical studies. To strengthen their conclusion, at least one experiment with CRC mice should be added to the revised manuscript.

Reviewer 2 Report

I have reviewed the paper by Qin et al.

The authors need to be more specific in which communities they are making they 2HB measurements, since they are working with colon cancer. “Gut bacteria” and “gut microbiota” needs to be replaced with colon microbiota.

Vague statements like “Previous work suggests that the liver and skin are significant sources of 2HB[15,16]. ” need to be leveled up.

The title should not use abbreviations and hydroxybutyric acid needs to be spelled. The title mentions mice, but the study actually handles human serum samples.

Authors need to explain what is blastp, and correct its spelling throughout the paper. At the end of the introduction is indicated that the bacteria cultured in 2.3, were identified through blastp (2.2). Overall the Methods are poorly in details of coherent flow and explanation.

It is obvious the Fusobacterium nucleatum is the main finding for this paper, yet it is not mentioned in the title. Figure 5 F shows a difference between 2HB and Fn and Fn+2AB, which is barely described in 3.5. This needs to be central in the Discussion, to correlate any mechanism in Colon cancer. Yet the authors are more concerned with what not found in serum

Reviewer 3 Report

In this study, the authors found a 1.74-fold increase in 2HB levels among patients with colorectal cancer (CRC). Additionally, this study provides a detailed profile of the bacteria that produce 2HB and highlights the significant contribution of gut microbiota to 2HB concentration in the intestinal lumen, although this contribution is relatively minor in serum 2HB concentration. Although the manuscript is well-structured and clear, the authors may wish to consider revising it to further enhance its overall quality.

1. The authors should specify the agreement number related to the ethics of human and animal research.

2. In this study, the authors used 400 human serum samples, including 198 healthy individuals and 202 CRC patients. However, it is unclear to the reviewer why this approach was chosen and how the results of these clinical specimens contribute to the overall findings of the manuscript. The authors are kindly requested to provide an explanation for the use of these human serum samples and to elaborate on their significance in the discussion section of the manuscript.

3. The content of the first and second paragraphs of the Discussion section seems to overlap with that of the Introduction. The reviewer recommends removing the redundant information to avoid repetition and improve the overall coherence of the manuscript.

4. As previously mentioned, the results from the 400 human serum samples appear to have limited relevance to the overall scope of the manuscript. Additionally, the authors did not conduct any gut microbiota testing on these patients. Therefore, it is recommended that the authors provide a detailed explanation in the discussion section regarding the relevance and limitations of this aspect of the study. Alternatively, the authors may consider removing this section to maintain the integrity and readability of the manuscript.

Round 2

Reviewer 1 Report

In the revised version of manuscript, authors addressed and fixed most of criticism. The manuscript can be accepted in present form.